# Trophic Niche Breadth of Falconidae Species Predicts Biomic Specialisation but Not Range Size

**DOI:** 10.3390/biology11040522

**Published:** 2022-03-29

**Authors:** Juan A. Fargallo, Juan Navarro-López, Juan L. Cantalapiedra, Jonathan S. Pelegrin, Manuel Hernández Fernández

**Affiliations:** 1Departamento de Ecología Evolutiva, Museo Nacional de Ciencias Naturales—Consejo Superior de Investigaciones Científicas, José Gutiérrez Abascal 2, 28006 Madrid, Spain; juannavarrolopez@gmail.com; 2Departamento de Ciencias de la Vida (Sección de Ecología), Facultad de Biología, Universidad de Alcalá, 28805 Alcalá de Henares, Spain; juan.lopezcantalapie@uah.es; 3Grupo de Investigación en Ecología y Conservación de la Biodiversidad (EcoBio), Área de Biología y Programa de Maestría en Educación Ambiental y Desarrollo Sostenible, Facultades de Educación y Ciencias Básicas, Universidad Santiago de Cali, Calle 5 62-00, Cali 760035, Colombia; jonathan.pelegrin00@usc.edu.co; 4Departamento de Geodinámica, Estratigrafía y Paleontología Facultad de Ciencias Geológicas, Universidad Complutense de Madrid, José Antonio Novais 12, 28040 Madrid, Spain; hdezfdez@ucm.es; 5Departamento de Cambio Medioambiental, Instituto de Geociencias (UCM, CSIC), Doctor Severo Ochoa 7, 28040 Madrid, Spain

**Keywords:** biome, ecological specialisation, diet richness, diversity, Falconiformes, generalist, phylogenetic signal, specialist, species distribution

## Abstract

**Simple Summary:**

The diversity of food consumed and the habitats occupied by species and populations, which is called a trophic niche, play a key role in biogeographic distribution patterns. Niche narrowing is considered a strategy to reduce resource-use overlap and competition among coexisting individuals, populations or species. However, the success of narrowing (specialism) or broadening (generalism) the trophic niche seems to depend on the temporal and spatial predictability of food resources. It is thought that specialist strategies are favoured in homogeneous environments predictable in time and space, while environmental unpredictability promotes generalist strategies that, in turn, allow a greater capacity to colonise new niches, thus tending to increase the geographic range size. Interestingly, although known at a within-population scale, the relationship between diet breadth and environmental heterogeneity at global and interspecific scales has not been explicitly assessed. Using the family Falconidae (Aves, Falconiformes) as a model study, we tested the prediction that those species with a wider diet spectrum will have larger geographic range sizes and inhabit more biomes. Our findings show that trophic breadth was not a good predictor for range size but was for total environmental heterogeneity, with more diet-generalist species occupying a higher number of biomes.

**Abstract:**

Trophic niche breadth plays a key role in biogeographic distribution patterns. Theory posits that generalist strategies are favoured in a more heterogeneous set of environments across a spatio-temporal gradient of resources predictability, conferring individuals and species a greater capacity for colonising new habitats and thus expanding their distribution area. Using the family Falconidae (Aves, Falconiformes) as a model study, we tested the prediction that those species with a wider diet spectrum will have larger geographic range sizes and inhabit more biomes. We assessed the relationships between trophic breadth (diet richness and diversity) at different taxonomic resolutions of the prey (class and order), range size and biomic specialisation index (BSI; number of biomes inhabited) for the different species. Despite different diet breadth indexes and taxonomic resolutions defined differently the trophic niche of the clade and species, our findings revealed that trophic breadth was not a good predictor for range size but was for total environmental heterogeneity, with more diet-generalist species occupying a higher number of biomes. Diet breadth at the order taxonomic level showed a higher capacity of predicting BSI than at class level, and can be an important ecological trait explaining biogeographic patterns of the species.

## 1. Introduction

Trophic niche breadth largely determines the global distribution patterns of species and influences their origin, extinction and diversification [1,2,3,4]. Resource competition among individuals of the same and different species is a major selective force shaping trophic niche and promoting ecological specialisation [2,4]. Niche narrowing or specialisation is considered a strategy to reduce resource-use overlap and competition among coexisting individuals, populations or species [5,6]. However, the success of narrowing or broadening the niche seems to depend on the temporal and spatial predictability of resources [7,8,9]. It is thought that a specialist strategy is favoured in predictable homogeneous environments in time and space [10,11]. Environmental unpredictability makes specialists more vulnerable to erratic variations of the resources exploited, and so these environments promote niche broadening or generalist strategies to facilitate rapid change in the exploitation of variable and unpredictable resources [10,11], allowing this flexibility a greater ability for colonizing new niches [12,13].

Island Biogeography Theory proclaims that larger areas contain a greater number of species because diversity of environments increases with area [14]. Thus, the increase in the number of environments would imply a mixture of environments with different characteristics, favouring the development of different species resource-use strategies. More heterogeneous environments may provide more diverse ways of exploiting the environmental resources (niches) allowing their exploitation by a greater number of species than the most homogenous ones (habitat heterogeneity hypothesis) [15,16]. On the other hand, from a biogeographic perspective, more stable biomes in terms of predictable temperature and precipitation, such as some in equatorial regions [17], are thought to boost the emergence and maintenance of a higher number of specialist species. In more variable biomes, where resources may have a more seasonal distribution, such as temperate forests compared to evergreen tropical forests [17], generalist species can be favoured, which may explain the latitudinal gradient of species richness widely observed in many taxa [3,18,19]. In addition, the greater colonising capacity favours the generalist species occupying a greater number of biomes and larger distribution areas [20]. Therefore, in general terms, the occupation of a more heterogeneous group of environments in space and time sets limits to the degree of species specialisation, promotes the organisms to increase breadth tolerance (plasticity) and favours polymorphism, genetic heterozygosity and trophic breadth at population and species levels [5,7,21,22,23,24,25].

Diet has been largely used as the basis for the study of ecological specialisation, as the availability of food resources is a major environmental force modulating adaptive strategies and phenotypes of the organisms and promoting intra- and interspecific competition [2]. Trophic breadth is usually considered a measure of adaptability [26,27], and thus, it can be predicted that diet generalist species will occupy larger ranges and more environments or that an increase in the geographical range size and environmental heterogeneity promotes trophic generalist strategies [28].

The relationship between the range size and diet breadth has been relatively well studied at the interspecific level, and although positively correlated in some herbivorous insects [29], no clear correlation between the two variables is observed in other clades [30]. A negative correlation has also been observed in some taxonomic groups, such as frog species of the genus *Cophixalus* [31], which suggests that other ecological factors, such as niche position (availability of environments and resources that are exploited [32]) or the scarcity of food resources in small restricted geographical areas (small islands, mountain tops) may force species and populations to widen the spectrum of food consumed in order to survive [31,33]. It is, therefore, necessary to carry out more studies in different taxonomic groups to achieve a more complete picture about the potential of trophic breadth to predict range size.

Knowing the relationship between diet breadth of species and the diversity of inhabited habitats, landscapes, biomes or environments is essential to understanding the concept of niche; however, it has rarely been studied. Within-population studies have reported that individuals occupying more fragmented habitats or more heterogeneous environments also show more plastic or generalist feeding behaviour reflected in a greater variety of food consumed [8,9,34,35,36], although the opposite has also been observed in other studies [37], which suggests that trophic adaptability can also be the result of active searching for different food items within the same habitat type. Therefore, trophic breadth may not necessarily predict the number of environments occupied. Surprisingly, as far as we know, the relationship between diet breadth and environmental diversity has not been explicitly studied at the interspecific and global scales. Some studies have reported that range size or latitudinal extent of the species correlates with both diet and environmental breadth, the latter measured as habitat heterogeneity, habitat variability, geographic mosaic or habitat specialisation [3,27,29,30,38], for which a positive correlation between diet breath and environmental amplitude is also inferred. In the same way, and mainly focused on herbivorous insects, the oscillation hypothesis proposes pulses of expansions (generalisation) and contractions (specialisation) of host-plant species consumption, speciation being resulted from the evolution of specialised populations derived from more generalist ones [24]. One of the bases of this idea is the positive correlation frequently found in this group between diet breadth (number of consumed host-plants) and geographic range size [24], as larger geographic ranges are likely to be more heterogeneous and include more species of host plants.

The aim of this work is to explore, at the interspecific level, whether diet breadth can predict diversity of inhabited environments and range size distribution. If a more generalist strategy, in terms of obtaining food resources, favours geographical expansion and settlement in new areas and habitats, trophic niche breadth should be positively correlated with diversity of occupied environments and range size at interspecific and global scale. We also analysed the potential influence of the index of trophic breadth chosen in relation to the taxonomic rank.

We used the family Falconidae as a study case because it is diverse (61 species in 9 genera), globally distributed, naturally occurring in all continents (except Antarctica), showing a great ecological diversity, with lineages adapted to all terrestrial biomes, and presenting a great variation in geographical range from species with restricted distributions to cosmopolitan species [39]. In addition, there is broad knowledge on diet in different populations of this group, so the taxonomic resolution of the diet and the magnitude of each type of prey can be refined in a large number of cases. Furthermore, the phylogenetic relationships of the family Falconidae are also well characterised [40], providing an opportunity to carry out interspecific comparisons within this taxonomic group. All these features make Falconidae a good model to study trophic breadth and allow us to analyse, for the first time, the relationship between diet, environmental diversity and range size at the global scale.

## 2. Methods

### 2.1. Trophic Niche Breadth

Information about methods used to describe diet was dissimilar among reviewed studies, with more general works (handbooks, atlas, etc.) reporting less detailed information. In several studies, the species, genus or family could not be identified for many prey items (mainly invertebrates), so the trophic niche breadth was determined using the order and class taxonomic levels, which are the levels that make the data comparable among different studies.

Diet richness (DR) was estimated using the number of prey found in 188 studies covering 466 studied populations in all 61 Falconidae species (Appendix B; Table A1; Appendix A). However, due to novel species reports after the publication date of specific diet study, we combine all information sources following the phylogeny proposed by Fuchs et al. [40]. This is the case for *Falco pelegrinoides* and *F. peregrinus*, which are not differentiated in diet studies [40], so that we worked only with *F. peregrinus*, assuming that they have a close phylogenetic relation. On the other hand, it should be noted that the studies used were carried out in different seasons of the year, although all were used equally in order to obtain a sufficient sample size. For each species, DR was considered as the total number of different prey taxa considering all the studies. We estimated DR at both class (DRc) and order (DRo) taxonomic rank (Appendix C; Table A2). We tried to minimise the sampling effort effect by controlling DR for the number of studied populations. The relationship followed the typical species accumulation curve (SAC) of type Y = a + b × X × c^−1^. Both DRc and DRo were significantly correlated with the number of studied populations (*R*^2^ = 0.38, *F*_1,59_ = 35.9, *p* < 0.001 and *R*^2^ = 0.72, *F*_1,59_ = 150.5, *p* < 0.001, respectively; Appendix A). We used the residuals as a way of detrending the DR with respect to the number of studies (dDRc and dDRo).

Diet diversity (DD) was calculated as the mean Shannon–Wiener index (SWI) of populations studied for each Falconidae species using PAST software [41]. We analysed diet diversity in 170 populations from 133 studies of 31 species at the class taxonomic rank (DDc) and 161 populations from 126 studies of 30 species at the order taxonomic rank (DDo; Appendix B and Appendix C). Thus, each DD value assigned to each species is the mean diversity calculated using all populations studied for that species. For this purpose, we used studies in which the number of prey taxa (86 studies) and/or percentage (116 studies) and/or biomass percentage (32 studies) was reported. Studies in which the fraction of undetermined prey taxa was higher than 30% were excluded from the analyses. Neither DDc nor DDo were affected by the number of studied populations (both *p* > 0.57; Appendix A).

Maximum diet diversity (MDD) was estimated as the highest SWI value found within all populations studied for a given Falconidae species, and also at the class (MDDc) and order (MDDo) taxonomic ranks. Both MDDc and MDDo were significantly higher when more populations were considered in the sample (*R*^2^ = 0.20, *F*_1,31_ = 7.9, *p* = 0.008 and *R*^2^ = 0.32, *F*_1,30_ = 14.3, *p* < 0.001, respectively; Appendix A). Therefore, we also used MDD residuals to detrend MDD (dMDDc and dMDDo) in order to control for sampling effort.

Carrion and vegetation are special food types that are difficult to classify as they have ecological characteristics that are very different from active living prey. Since the aim of estimating diet richness/diversity using taxonomic groups of the prey is analysing the breadth of niches consumers must explore to obtain food, carrion and vegetation were similarly included in the models as different prey groups, as it has been used in other diet studies [42,43,44,45]. In this study, we followed this procedure by considering carrion and vegetation as two different classes or orders when present. Lacertillia, Serpentes, Iguania and Gekkota, were considered as separate orders [46].

### 2.2. Geographical Range Size and Biomic Specialisation

The geographic range size was estimated using distribution maps published in the Handbook of the Birds of the World [39], representing the extent of occurrence for each species. The total area was calculated in these maps through “wand tools” in Image J software [47,48]. Maps were scaled using the distance of the Equator (latitude 0°) and the Tropic of Capricorn (23°27′ S) across the continents. The extent of occurrence areas used included sedentary, wintering and breeding ranges when applicable.

Regarding the diversity of inhabited environments, most studies have analysed heterogeneity in land cover, topography and vegetation, while few studies used climatic or soil environmental heterogeneity [49]. In this sense, biomes are spatial units that contain their own environmental characteristics, and also in terms of temperature and precipitation [17], the two main variables defining biomes classifications [50]. The biomic specialisation index (BSI, number of biomes inhabited by a given species) [20] may be considered as an indicator of the environmental heterogeneity that a species must deal with at the global scale, since it describes the position of a species in the specialist–generalist gradient through the capacity of the species for obtaining resources in different environments [20,51].

The biomic specialisation index (BSI) was estimated as the number of biomes inhabited by a given species. Walter’s classification of biomes [40] is based on the regional vegetation and the annual climatic interaction between precipitation and temperature. This typology was originally chosen for a measure of ecological specialisation [19] because of its simplicity and concordance with traditionally recognized major terrestrial biomes on Earth. Only ten different climate zones, placed in only one hierarchic level, allow us to differentiate the main biomes in the world. On the contrary, most other climatic classifications differentiate many more biomes, which may hamper the statistical power of any analysis, or very few, which does not allow to properly differentiate species ecology within such loosely defined biomes. BSI was determined following the procedure described by Hernández Fernández [52] and Hernández Fernández and Vrba [19] to determine the number of biomes (also known as zonobiomes or climate zones) inhabited by each species [51,53]. Briefly, if 15% or more of the geographical range of a species is situated within a biome, the species was recorded as present in that biome. As some climatic dominions are small enough to comprise less than 15% of the total distribution ranges of species with large range sizes, a species was also recorded as present in a specific biome if it inhabits 50% or more of one climatic dominion. A climatic dominion was defined as a continuous terrestrial area within one biome only [52]. For instance, the Zaire basin is a climatic dominion of the evergreen tropical rainforest biome and it is geographically separated from the Ivory Coast, another African climatic dominion of that type of biome. We also considered those species inhabiting mountainous ranges as adapted to the biomes represented by analogous habitat series in the altitudinal gradients, since these habitats present very similar vegetation physiognomy and ecological pressures. Therefore, the presence in a mountain vegetation belt was also recorded as presence in the corresponding analogous biome [19]. BSI equals 1 for most specialised species, whereas for generalist species it could be as high as 10.

### 2.3. Statistical Procedures

Data were analysed using phylogenetic generalised least squares (PGLS) regressions. We followed the recent Falconidae phylogeny [43]. To analyse the relationship between geographic range size and trophic niche breadth, we carried out PGLS regressions for each taxonomic rank. In these analyses, range size was included as the dependent variable and diet variables (dDR, DD and dMDD) as the independent variables. Since range size did not show a normal distribution, this variable was log-transformed (Kolmogorov–Smirnov test *d* = 0.12, *p* = 0.36) when residuals of the models were not normal. To analyse the relationship between biomic specialisation and trophic niche breadth, we carried out PGLS regressions with BSI (interval variable) as the dependent variable, and DR, DD and MDD as explanatory variables. Since species occupying larger distribution areas are expected to inhabit a higher number of biomes (higher BSI), the area of the geographical distribution was also included as a covariate in these models. BSI did not show a normal distribution, for which log-transformed BSI (Kolmogorov–Smirnov test *d* = 0.14, *p* = 0.14) was used when residuals of the models were not normal.

Since it could be expected that migratory species have larger range size and inhabit more biomes than sedentary ones, the migratory status of the birds was also considered as a cofactor in a preliminary analysis. To this end, a migratory species was defined as any species that has a wintering or breeding region, or both, in addition to the sedentary region (Appendix B; Table A1).

Diet variables were highly intercorrelated, showing elevated variance inflation factors (all VIFs > 2; Appendix A) when included together in a model. Therefore, to avoid bias due to multicollinearity in the analyses [54], we studied the statistical effect of each trophic variable on range size and BSI in separate regression models. Pagel’s Lambda was estimated to test for phylogenetic signal [55] by maximum likelihood in all PGLS regressions. All analyses were performed using R software, version 2.14.2 [cran, 2012] and the R package caper [cran, 2013; [56]] (Appendix A). Mean ± S.D. values are shown.

## 3. Results

### 3.1. Diet Richness and Diversity

The prey class most frequently consumed by Falconidae was Aves, consumed by 93.6% of species, followed by Insecta (87.1%), Mammalia (77.4%) and Reptilia (75.8%). However, at the order rank, the most frequently consumed was Lacertilia, consumed by 72.6% of species, followed by Passeriformes (59.6%), Rodentia (58.1%) and Orthoptera (46.8%).

Mean DRc was 5.8 ± 2.5, ranging from 1 to 13 (Appendix C; Table A2), and its frequency distribution was slightly skewed to the right (skewness = 0.91; Figure 1). Mean DRo was 11.1 ± 9.8, ranging from 1 to 43 (Appendix C; Table A2), with its frequency distribution clearly skewed to the right (skewness = 1.6; Figure 1). The southern crested caracara *Caracara plancus* showed the widest trophic breadth (13 classes and 43 orders), while the slaty-backed forest falcon *Micrastur mirandollei* showed the narrowest one (1 class and 1 order; Appendix C; Table A2).

DDc was 0.6 ± 0.4 ranging from 0 to 1.28 (Appendix C; Table A2) and skewness = 0.3 (Figure 1). DDo was 1.1 ± 0.5 ranging from 0 to 2.0 (Appendix C; Table A2) and skewness = −0.30 (Figure 1). MDDc was 0.8 ± 0.4 and skewness = 0.0 (Figure 1). MDDo was 1.4 ± 0.5 and skewness = −0.81 (Figure 1).

### 3.2. Geographical Range Size and Biomic Specialisation Index

Mean BSI of the Falconidae group was 3.3 ± 2.2 ranging from 1 to 10 biomes (Figure 2). Fifteen Falconidae species (24.59%) inhabit only one biome, 33 species (54.10%) two to four and 13 species (21.31%) inhabit five or more biomes (Figure 2). One species, the peregrine falcon *Falco peregrinus* inhabits all 10 terrestrial biomes. The frequency distribution of BSI was skewed to the right (skewness = 0.96; Figure 2).

The most frequently occupied biome (Appendix A) was the tropical deciduous woodland (II) with 38 species, followed by savannah (II/III) with 26 species, while the least inhabited biome was the tundra (IX) with only five species. The biome with the most specialist species was the evergreen tropical rainforest (I), with five species exclusively occupying this biome. *Caracara* (one species) was the most generalist genus (BSI = 9), and *Falco* (39 species) was the genus with the second largest BSI (3.7 ± 2.4). On the other hand, *Polihierax* and *Spiziapteryx* were the most specialist genera (both BSI = 1).

### 3.3. Relationships among Diet, Geographical Range and Biomic Specialisation

In relation to range size, migratory status had no significant effect in any of the analyses (all *p* > 0.29), so it was excluded from the PGLS models. PLGS regression models showed that the range size of Falconidae was not significantly explained by dDR, DD or dMDD (Table 1). Furthermore, no clear phylogenetic influence was observed, as values of lambda were close to zero for all models (Table 1).

In relation to BSI, migratory status had no significant effect in any of the analyses (all *p* > 0.20), so it was excluded from the PGLS models. PGLS regression models showed a significant positive correlation between BSI and the geographic range size (Table 2; Figure 3); that is, those species with larger range sizes also occupy more biomes. Controlling for geographic range size, PGLS regressions showed that species with a higher dDRo, but not dDRc, were also present in more biomes (Table 2; Figure 3).

In the case of DD, the models also showed a significant and positive correlation of BSI with DDo, but not with DDc (Table 2; Figure 3). For MDD-res, BSI correlated marginally significantly and positively with both dMDDc and dMDDo (Table 2; Figure 3). Phylogenetic influence was virtually nil in all models, as indicated by lambda being close to zero (Table 2).

## 4. Discussion

### 4.1. Trophic Niche Breadth

A remarkable first inference that can be drawn from our study is the influence of the diet index chosen on the description of trophic breadth of the clade. Using diet richness, the distribution frequency of Falconidae is markedly skewed to the left, which describes a group that tends to specialise, particularly when richness is estimated at the order taxonomic rank. However, using diet diversity, the group shows a more centred distribution of frequencies with no clear trend to specialism or generalism. This is due to the shift of the frequency distribution to the right when richness is controlled for evenness, that is, diversity, since the values assigned to more specialised species increase with respect to more generalist species. In other words, many of the most specialist species (36% and 60% of the species consume a maximum of 5 and 10 orders, respectively) show a high balanced consumption of the prey orders, whereas this does not seem to occur in the more generalist species, indicating that many of the orders consumed by the species at this extreme of the distribution show low relevance in the diet. Diet richness gives equal importance to both the occasional and preferential consumption of food types [57], which overestimates the generalist capability and the number of generalist species and heavily depends on sampling effort [58]. Richness and diversity provide different information, in the case of diet, data extracted from studies conducted in specific populations, the diversity index can provide a more realistic view because it decreases the weight of irrelevant or anecdotal food items found in local populations.

Taxonomic resolution also influences the description made of the trophic niche defining the group. At the order rank, the Falconidae family can be defined as mainly lizard consumers (Lacertilia), although passerine (Passeriformes) and rodent species are highly represented in the diet followed in importance by grasshoppers (Orthoptera). However, at the class rank, Falconidae is mainly composed of bird-consumer species, with Insecta, Mammalia and Reptilia classes having a great representation in the diet as well. Differentiation of prey species at class or order ranks affects the trophic breadth calculated for each species. Clear examples for this discrepancy are the peregrine falcon (*Falco peregrinus*) and the Eleonora’s falcon (*Falco eleonorae*), two bird-specialist consumers showing low values of diet richness and diet diversity (closer to specialism) when working at the class rank, but showing much higher values (closer to generalism) when estimating the trophic breadth at the order rank. Resource species share functional, habitat, spatial or ecological characteristics that can favour or hamper their consumption by a given consumer species [59,60,61], therefore influencing the trophic specialisation category assigned to the species [60]. Our study highlights the importance of the measure of trophic breadth selected to describe and analyse the niche of organisms, as the description for a given taxonomical groups can differ markedly.

### 4.2. Range Size

Our study shows that the trophic breath measured as diet richness or diet diversity does not predict range size of Falconidae species, which coincides with what has been found in general for different taxonomic groups [30]. The niche of a species is determined by many ecological factors. Some of them, such as niche position, could have a greater relevance than trophic breadth to predict how widespread a species is [32]. In this sense, our study shows a new clade that adds to the list of clades in which no clear relationship between geographical range size and diet has been found. In addition, our results also suggest that in the case of the Falconidae group, more generalist species may not have a greater capacity or need for spatial expansion than more specialists. Similar results have already been found in African large mammals, which show a polygonal relationship between range size and biome, indicating that species occupying many biomes tend to have a large range size, while the range size of species occupying few biomes may strongly vary depending on the geographical size of the biomes they occupy [20].

### 4.3. Biomic Specialisation

The frequency distribution of the BSI for the family Falconidae showed a high number of species using only a few biomes, similarly to other taxonomic groups [20,51,53], which coincides with the high representation of specialist species in nature [62,63]. The most frequently inhabited terrestrial biome by Falconidae species is the tropical deciduous woodland. This result is similar to that found by Belmaker et al. [3], who concluded that the geographical area occupied by this biome showed the highest richness in bird species. Latitudinal gradients found for species richness and speciation also place its maximum in this tropical biome [32]. Furthermore, the highest number of biomic specialists (BSI = 1) was found in another tropical biome, the evergreen tropical rainforest, containing 20% of biomic specialist species. These results are in agreement with the idea that since such environments favour specialisation [3], more predictable environments are home to more specialist species than would be expected.

Range size and biome occupation were estimated using extent of occurrence areas, for which included-but-unoccupied habitats and areas were considered in the estimations. Our study was conceived on the premise that this potential problem affects all species randomly; therefore, species that occupy larger areas, for whatever reason, could occupy more biomes. To control for this possible effect, the BSI was analysed by including the range size as a covariate in the model. Despite not being able to control the potential variation due to included-but-unoccupied habitats, the results show significant correlations between BSI and most diet breath indexes at order taxonomic level.

When controlling for range size, our results show that trophic breadth can predict BSI, as those Falconidae species showing a higher trophic breadth, at least in terms of diet richness and diversity at the order level, are also capable of living in more biomes. A marginally significant trend (explaining 39% (class) and 37% (order) of the variance) was also observed when using the maximum diet diversity, an index that we consider measures the potential flexibility of the species to broaden the feeding spectrum. Models for diet richness and diet diversity were significant only when diet is analysed at the order rank, probably because order shows a greater variance in the diet and the predilection of the species for some prey orders may be uneven within the same class.

## 5. Conclusions

Our study indicates that while trophic breadth seems to show little capacity for predicting the geographic range size of the species, on the contrary, it has a great potential to predict the number of inhabited ecosystems. At the interspecific level and on a global scale, this finding provides support to the idea that a generalist trophic strategy allows the species a greater plasticity to inhabit a greater variety of environments, which has already been found at within-population scales. However, the correlative nature of our study prevents us from knowing causality, so our results could be explained by stating that species with a greater capacity to occupy different biomes (or another characteristic not measured in this study) also have a broader trophic niche. Additionally, our study shows that the choice of the trophic breadth index, and the taxonomic resolution at which it is estimated, affect the categorisation made regarding the trophic niche of clades and the value of trophic niche breadth as a predictor of global biogeographic distribution patterns.

## Figures and Tables

**Figure 1 biology-11-00522-f001:**
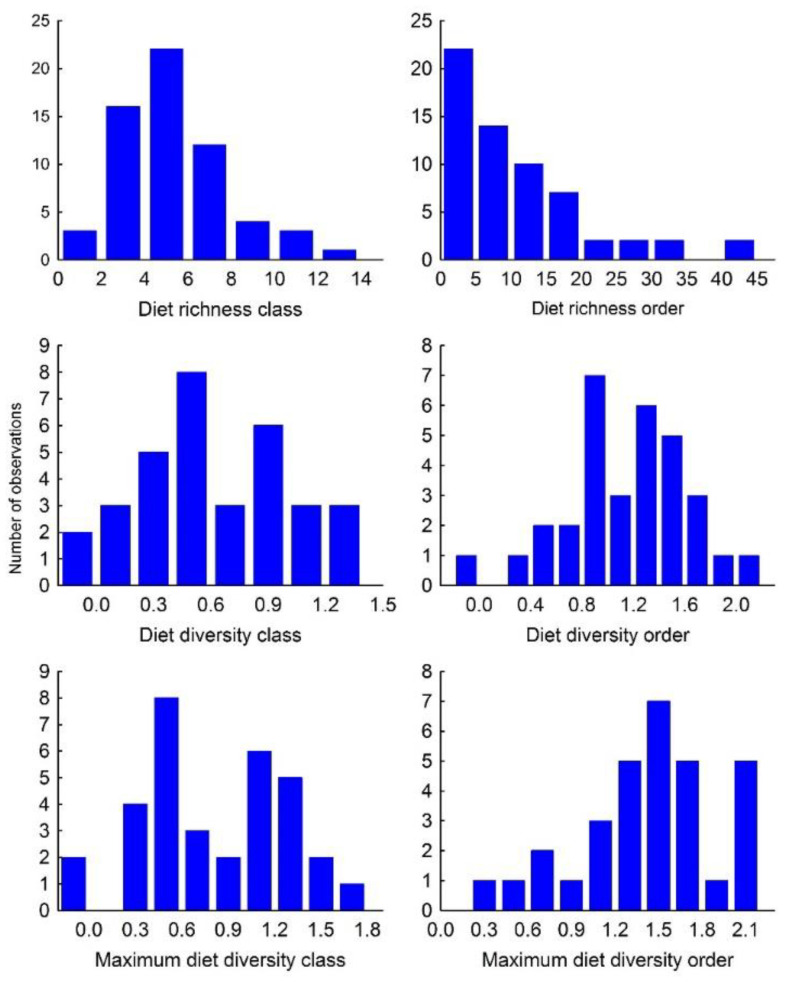
Frequency distribution of diet richness, diet diversity and maximum diet diversity at class (**left**) and order (**right**) taxonomic ranks of Falconidae species.

**Figure 2 biology-11-00522-f002:**
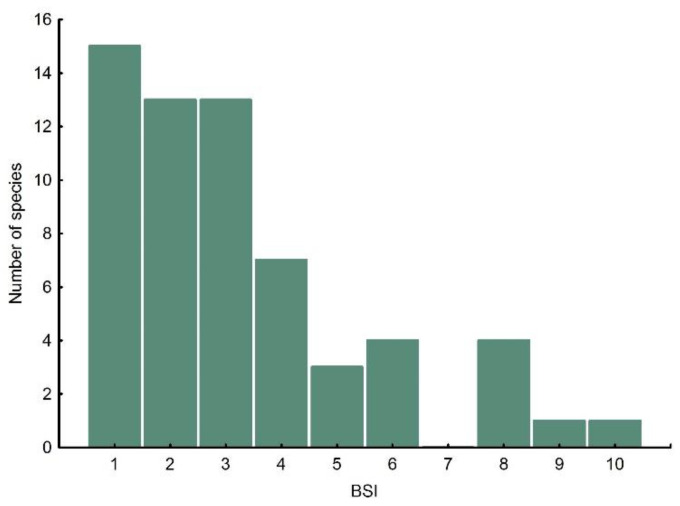
Frequency distribution of the biomic specialisation index (BSI) in the Falconidae species.

**Figure 3 biology-11-00522-f003:**
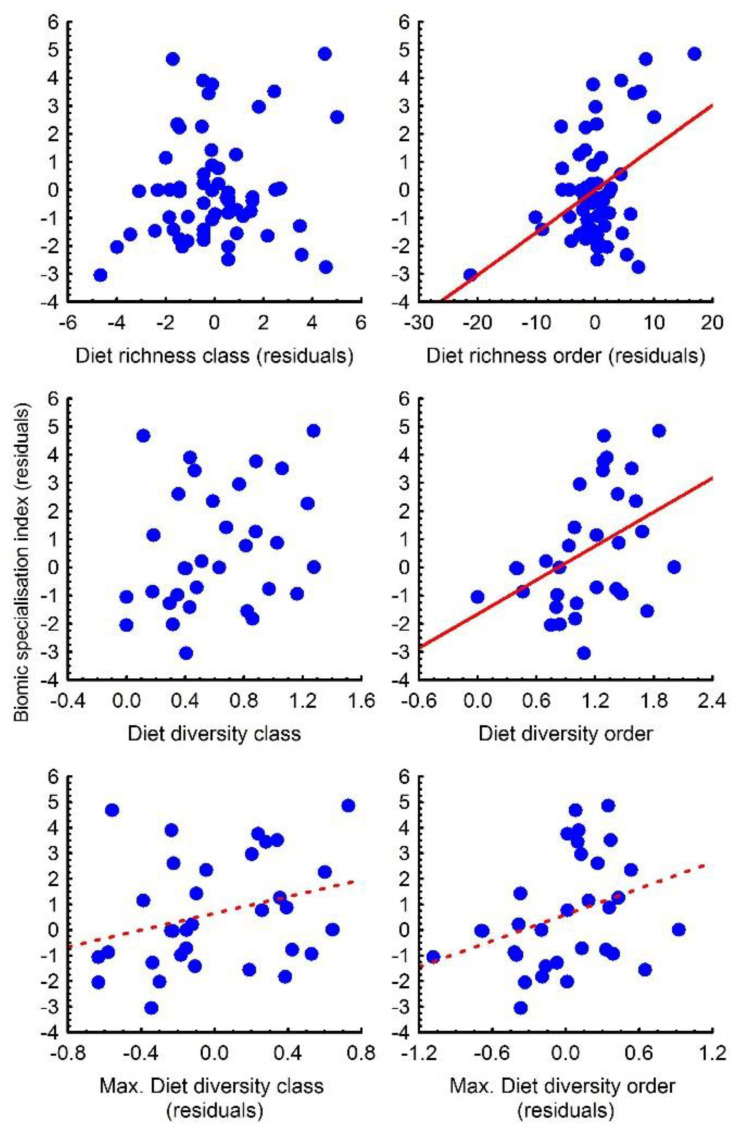
Relationships between biomic specialisation index (BSI) controlled for range size (residuals) of Falconidae species and the different diet indexes of trophic breadth (richness, diversity and maximum diversity) estimated at class and order taxonomic ranks. Diet richness and maximum diet diversity were detrended (residuals) with respect the number of studied populations. Dashed lines represent marginally significant correlation (*p* values between 0.06 and 0.09).

**Table 1 biology-11-00522-t001:** Results of phylogenetic generalised least squares (PGLS) models for the relationship between geographical range size and detrended diet richness residuals (dDR), diet diversity (DD) and detrended maximum diet diversity (dMDD) at class (c) and order (o) taxonomic ranks of Falconidae species.

Effect	R*^2^*	λ	Estimate	*SE*	*t*	*F*	*p*	*n (d.f.)*
dDRc	0.02	<0.001	0.023	0.15	0.16	0.02	0.876	61 (2,59)
dDRo	0.02	<0.001	0.015	0.06	0.28	0.08	0.781	61 (2,59)
DDc	0.07	<0.001	−2.084	1.295	−1.609	2.59	0.118	33 (2,31)
DDo	0.01	<0.001	0.041	1.118	0.036	0.01	0.971	32 (2,30)
dMDDc	0.04	<0.001	−1.858	1.23	−1.51	2.28	0.141	33 (2,31)
dMDDo	0.03	<0.001	0.076	1.20	0.06	0.01	0.951	32 (2,30)

**Table 2 biology-11-00522-t002:** Results of phylogenetic generalised least squares (PGLS) models for the relationship between biomic specialisation index (BSI) and detrended diet richness (dDR), diet diversity (DD) and detrended maximum diet diversity (dMDD) at class (c) and order (o) taxonomic ranks of Falconidae species. Data on raw models and after the inclusion of range size of the distribution area as a covariate are shown. *R^2^* and *F* values correspond to the whole model and estimate, *t* and *p* values to the independent variable. The effect of range size was significant (all *p* < 0.001) in all models.

Effect	R*^2^*	λ	Estimate	*SE*	*t*	*F*	*p*	*n (d.f.)*
dDRc	0.01	<0.001	0.039	0.01	0.88	0.76	0.385	61 (2,59)
dDRc + Range size	0.36	<0.001	0.035	0.03	0.97	17.57	0.334	61 (3,58)
dDRo	0.12	<0.001	0.161	0.05	3.04	9.24	0.004	61 (2,59)
dDRo + Range size	0.41	<0.001	0.152	0.04	3.52	22.23	<0.001	61 (3,58)
DDc	0.02	<0.001	0.412	1.29	0.32	0.10	0.751	33 (2,31)
DDc + Range size	0.35	<0.001	1.701	1.07	1.59	9.46	0.123	33 (3,30)
DDo	0.09	<0.001	2.036	0.99	2.04	4.16	0.050	32 (2,30)
DDo + Range size	0.42	<0.001	2.013	0.80	2.25	12.12	0.018	32 (3,29)
dMDDc	0.02	<0.001	0.602	1.22	0.49	0.25	0.625	33 (2,31)
dMDDc + Range size	0.39	<0.001	1.753	0.99	1.75	9.88	0.089	33 (3,30)
dMDDo	0.05	<0.001	1.746	1.09	1.59	2.53	0.122	32 (2,30)
dMDDo + Range size	0.37	<0.001	1.704	0.89	1.91	10.09	0.066	32 (3,29)

## Data Availability

The data are shown in Appendix B and Appendix C.

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
