# Peer review of "Trophic Niche Breadth of Falconidae Species Predicts Biomic Specialisation but Not Range Size"

_biology, 2022, doi:10.3390/biology11040522_

Round 1

Reviewer 1 Report

This study is very interesting, and is generally well written and clearly presented. The scientific background that justifies the questions of the study are clear. Some clarifications in the methods and results are necessary, but overall, methods are justified and applied correctly. Small issues that need further addressing are in specific comments below.

Good review!

Comments:

L 17. Please refer for which level of organization the diversity and habitats occupied play an important role. I think something like “…habitats occupied by species” or “…habitats occupied by populations”, or something similar.

L93.” The negative correlation between range size and diet breadth”

L93. Please add examples of those taxonomic groups which had negative correlation between range size and niche breadth, and what are the taxonomic levels used on those studies, order, family, species?

L97 to L 99. I don´t see how your previous sentence takes to this conclusion. I suggest modifying or deleting this sentence.

L100, please clarify what “a specific way” means. Did you mean at specific levels?

L127. Start the line in “Most studies analyzed…” and end the sentence by citing the paper [39].

L129. Maybe a new paragraph could start in “Biomes…”

L 166. “methods used to describe diet” instead of “diet description procedures”

L 168. “Studies” instead of “Studios”, and please cite examples of those studies

L 174. “of” instead of “for”

  1. 177. “they have a close phylogenetic relation” instead of “are phylogenetically closely related”. So, this means you used in some cases a complex of species instead of a single species. Did you used the joined distribution range of the species as well? Can you please indicate which species represent a complex of two or more species (in the appendix maybe)?

L185. DR increased with the number of studies, or each study evaluated single populations? Furthermore, legend of figure S1 needs to have more information. Check labels for typos.

L 187. Could you say those residuals represent detrended DR? maybe would be easier to call them dDRc and dDRo

L191 to 194. So, in Figure S1, each point is a population? It needs to be clearer, as I was thinking that analysis were done in the species level, and this section gave the impression that it was done at the population level (so, more than one sampling unit per species, or a single sampling unit per species/species complex)

L203 same comment as L187

L237 to 249. DR, DD and MDD are detrended right? Might be useful clarify.

L251. A table of VIF in the supplemental material would be useful.

L256. Would be very useful to have the R-scripts appended as well.

L 281. Should it be “4 or more biomes” instead of “8 or more biomes…”

L 343. I get the impression that availability of resource could also  affect the skewness, as generalists would often consume prey that are more available. Maybe it is worth to discuss about it?

Author Response

Comments and Suggestions for Authors

REVIEWER 1: This study is very interesting, and is generally well written and clearly presented. The scientific background that justifies the questions of the study are clear. Some clarifications in the methods and results are necessary, but overall, methods are justified and applied correctly. Small issues that need further addressing are in specific comments below.

Good review!

AUTHORS: Thank you very much for the comments.

Comments:

REVIEWER 1: L 17. Please refer for which level of organization the diversity and habitats occupied play an important role. I think something like “…habitats occupied by species” or “…habitats occupied by populations”, or something similar.

AUTHORS: Added.

REVIEWER 1: L93.” The negative correlation between range size and diet breadth”

AUTHORS: The correlation tends to be positive in general, but it does not always work out. We have specified it in the new version of the manuscript.

REVIEWER 1: L93. Please add examples of those taxonomic groups which had negative correlation between range size and niche breadth, and what are the taxonomic levels used on those studies, order, family, species?

AUTHORS: Added.

REVIEWER 1: L97 to L 99. I don´t see how your previous sentence takes to this conclusion. I suggest modifying or deleting this sentence.

AUTHORS: The key is the range restriction. Those species or populations with low colonizing capacity inhabiting low-productivity environments such as small islands or mountain tops (small range size) are “forced” to increase trophic breadth in order to survive. This has been observed in the examples quoted. Now, we have tried to explain it better. Thank you.

REVIEWER 1: L100, please clarify what “a specific way” means. Did you mean at specific levels?

AUTHORS: Removed.

REVIEWER 1: L127. Start the line in “Most studies analyzed…” and end the sentence by citing the paper [39].

AUTHORS: Done.

REVIEWER 1: L129. Maybe a new paragraph could start in “Biomes…”

AUTHORS: Done.

REVIEWER 1: L 166. “methods used to describe diet” instead of “diet description procedures”

AUTHORS: Changed.

REVIEWER 1: L 168. “Studies” instead of “Studios”, and please cite examples of those studies

AUTHORS: Changed,

REVIEWER 1: L 174. “of” instead of “for”

AUTHORS: Changed.

REVIEWER 1: 177. 177. “they have a close phylogenetic relation” instead of “are phylogenetically closely related”. So, this means you used in some cases a complex of species instead of a single species. Did you used the joined distribution range of the species as well? Can you please indicate which species represent a complex of two or more species (in the appendix maybe)?

AUTHORS: Changed. We are not sure if it can be considered as a complex of species, some authors simply call the pelegrinoides as peregrinus and others separate it. This has now been indicated in both appendices, as proposed.

 REVIEWER 1: L185. DR increased with the number of studies, or each study evaluated single populations? Furthermore, legend of figure S1 needs to have more information. Check labels for typos.

AUTHORS: DR increases with the number of studied populations as some studies evaluated more than one population. This has been clarified now. More information has been added to the legend of Fig. S1 and labels has been checked for typos.

REVIEWER 1: L 187. Could you say those residuals represent detrended DR? maybe would be easier to call them dDRc and dDRo

AUTHORS: Computing the residuals is one way to detrend a variable, so we think either way works. Anyway, we changed it, as proposed.

REVIEWER 1: L191 to 194. So, in Figure S1, each point is a population? It needs to be clearer, as I was thinking that analysis were done in the species level, and this section gave the impression that it was done at the population level (so, more than one sampling unit per species, or a single sampling unit per species/species complex)

AUTHORS: We say in the text “Diet diversity (DD) was calculated as the mean Shannon-Wiener Index (SWI) of populations for each Falconidae species”, that is, each point represents the mean diversity value of all the studied populations for each species. In the new version we have changed it so that there is no confusion. Thank you.

REVIEWER 1: L203 same comment as L187

AUTHORS: Changed.

REVIEWER 1: L237 to 249. DR, DD and MDD are detrended right? Might be useful clarify.

AUTHORS: Yes, DR and MDD were detrended. Changed. Thanks.

REVIEWER 1: L251. A table of VIF in the supplemental material would be useful.

AUTHORS: Included.

REVIEWER 1: L256. Would be very useful to have the R-scripts appended as well.

AUTHORS: Included.

REVIEWER 1: L 281. Should it be “4 or more biomes” instead of “8 or more biomes…”

AUTHORS: No, it's correct as we have it, but it's probably not the most intuitive way to describe the results. Now we have described it a little better. Thanks.

REVIEWER 1: L 343. I get the impression that availability of resource could also affect the skewness, as generalists would often consume prey that are more available. Maybe it is worth to discuss about it?

AUTHORS: We think this is why we expect more generalist species occupying more heterogeneous environments as more different environments offer higher availability of resources, thus affecting the skewness. This is the basis of the study. We do not know if we are responding correctly to the comment, we do not understand the idea well.

Reviewer 2 Report

Well written manuscript which shows interesting results. I think the methods used are correct, I only have a few minor suggestions for improvement.

L. 167-168, "In most studios" change to "studies"

L. 265. "5.8 ± 2.5" What does 2.5 indicate? I guess, SE, SD or something else, please add this throughout the manuscript.

L. 419-420. Remove sentence.

Author Response

AUTHORS: Thank you very much for the comments.

Comments and Suggestions for Authors

Well written manuscript which shows interesting results. I think the methods used are correct, I only have a few minor suggestions for improvement.

AUTHORS: Thank you very much for the comments about our study.

REVIEWER 2: L. 167-168, "In most studios" change to "studies"

REVIEWER 2: L. 265. "5.8 ± 2.5" What does 2.5 indicate? I guess, SE, SD or something else, please add this throughout the manuscript.

REVIEWER 2: L. 419-420. Remove sentence.

AUTHORS: All suggestions made have been followed. Thanks. A phrase at the end of “Statistical procedures” has been added to indicate that those values are mean ± SD.

Reviewer 3 Report

The manuscript titled “Trophic niche breadth of Falconidae species predicts biomic specialisation but not range size” (ID: biology-1610510).  Fargallo et al. present a survey on relations between niche breadth of falcons with their biogeographic specialization and distribution over the world. The general idea of this study is interesting, but not novel (numerous other studies related to the topic are presented in the introduction). The statistical methods chosen for this study are complex, interesting and appropriate, however, my major doubts about this study are about some basics of data collection and selection. First of all, this study strongly depends on data availability, and Appendix A shows that “sampling” is strongly biased by overrepresentation of available data for some taxa (majority of data are for two genera: Falco and Daptrius), whereas for most of other there are no data (e.g. Microhierax and Micrastur, and others), or data are very limited. This problem is even greater if considering the number of populations, for which diet was determined. It could be expected that if some taxa were examined only for single populations, diet breadth is probably underestimated, compared to widely studied species. Another serious problem with this study is that it neglects the history of species. Due to past climate and environmental changes, the diversity of falcons on higher latitudes is lower than in areas closer to the equator (this is a common pattern for biodiversity). The history (area of origin, post-glacial expansion etc.) probably also affected range sizes, as species in sub/tropics had a long time to speciate and partition their niches, when temperate, boreal or arctic taxa. However, the latter species often have wider ranges as they spread along with wider biomes (e.g. temperate or boreal forests or steppes of Eurasia and North America). So, the question of this study could be rather reverse – if biomic specialization/range size predicts niche breadth. I am not surprised that the Authors failed to find support for their hypothesis that trophic breadth was not a good predictor for range size. Finding that more diet-generalist species occupy a higher number of biomes is rather obvious.

I have also some specific comments:

The introduction is interesting and exhaustively describe the topic, but it is far too long, which make reading this hard – after 3-4 sections, the reader could lost in so much information. The introduction should be informative but also concise.

Methods

Maybe I misunderstood this, but for some species, data were available and used for their breeding, wintering and migration areas, whereas for others only for e.g. breeding sites? If yes, this is also a problem as migratory species have probably wider niche breadth as they need to hunt in various parts of the world, where available preys could differ and also the behaviour of birds could change (when feeding chicks, they could need to hunt on other preys than when had to find food for their own).

At the end of Conclusions is the strange sentence:

“This section is not mandatory but can be added to the manuscript if the discussion is 419 unusually long or complex.” – I suppose that is just some editorial error?

Why Appendix B is divided into 2 parts?

Walter & Box 1976 cited in Table S1 is missing in references. In manuscript is another reference: Fernández, M.H.; Morales, J. Biomic specialization and speciation rates.

There are other classifications of biomes, with some more recent and including more units. Authors should justify why they decide to follow that old classification?

Author Response

AUTHORS: Thank you very much for the comments.

REVIEWER 3: 1) The manuscript titled “Trophic niche breadth of Falconidae species predicts biomic specialisation but not range size” (ID: biology-1610510).  Fargallo et al. present a survey on relations between niche breadth of falcons with their biogeographic specialization and distribution over the world. The general idea of this study is interesting, but not novel (numerous other studies related to the topic are presented in the introduction).

AUTHORS: As we said in the introduction, none of those studies, nor other existing ones, explicitly analyse the relationship between the breadth of the diet and environmental heterogeneity at the inter-specific level.

2) The statistical methods chosen for this study are complex, interesting and appropriate, however, my major doubts about this study are about some basics of data collection and selection. First of all, this study strongly depends on data availability, and Appendix A shows that “sampling” is strongly biased by overrepresentation of available data for some taxa (majority of data are for two genera: Falco and Daptrius), whereas for most of other there are no data (e.g. Microhierax and Micrastur, and others), or data are very limited. This problem is even greater if considering the number of populations, for which diet was determined. It could be expected that if some taxa were examined only for single populations, diet breadth is probably underestimated, compared to widely studied species.

AUTHORS: We agree that this is a problem, but believe that it has been statistically addressed in our study by controlling diet variables by the number of populations from which data have been drawn, where necessary.

3) Another serious problem with this study is that it neglects the history of species. Due to past climate and environmental changes, the diversity of falcons on higher latitudes is lower than in areas closer to the equator (this is a common pattern for biodiversity). The history (area of origin, post-glacial expansion etc.) probably also affected range sizes, as species in sub/tropics had a long time to speciate and partition their niches, when temperate, boreal or arctic taxa. However, the latter species often have wider ranges as they spread along with wider biomes (e.g. temperate or boreal forests or steppes of Eurasia and North America). So, the question of this study could be rather reverse – if biomic specialization/range size predicts niche breadth. I am not surprised that the Authors failed to find support for their hypothesis that trophic breadth was not a good predictor for range size.

AUTHORS: We do not believe that this is a serious problem, at most it would be an idea to discuss our results. However, our results do not support the idea since species that have larger ranges also occupy a larger number of biomes, as we show in our study. We are not saying that it cannot be, since it seems a fairly intuitive idea, but our data does not corroborate it in this taxonomic group. And yes, we agree that range size and environmental heterogeneity may both be the explanation for diet breadth. We cannot know at an interspecific level what the cause-consequence is, very possibly in some species the diet is the cause of the expansion (range size and new habitat colonization) and in others it is the consequence. Our goal is to analyse whether diet can predict both biogeographical variables using a theoretical framework that allows us to make such a prediction and trying to highlight the importance of diet studies in explaining global patterns.

4) Finding that more diet-generalist species occupy a higher number of biomes is rather obvious.

AUTHORS: We also agree with this assessment. That is why it is so surprising that there are no studies that have directly analysed the relationship between diet breadth and environmental heterogeneity at interspecific and global levels.

I have also some specific comments:

REVIEWER 3: The introduction is interesting and exhaustively describe the topic, but it is far too long, which make reading this hard – after 3-4 sections, the reader could lost in so much information. The introduction should be informative but also concise.

AUTHORS: Changed. Thanks.

Methods

REVIEWER 3: Maybe I misunderstood this, but for some species, data were available and used for their breeding, wintering and migration areas, whereas for others only for e.g. breeding sites? If yes, this is also a problem as migratory species have probably wider niche breadth as they need to hunt in various parts of the world, where available preys could differ and also the behaviour of birds could change (when feeding chicks, they could need to hunt on other preys than when had to find food for their own).

AUTHORS: Migratory status has been included in the analyses in the new version (see appendix A). Migratory status had no significant effect in any of the analyses, neither in those of the range size nor in those of the BSI (see p values below), so it was excluded from the models.

Range size

dDRc

0.35013852

dDRo

0.33565421

DDc

0.41698643

DDo

0.29465869

dMDDc

0.8431899

dMDDo

0.86824144

BSI

dDRc

0.985556405

dDRo

0.201205881

DDc

0.478766201

DDo

0.456405712

dMDDc

0.525302643

dMDDo

0.852205398

REVIEWER 3: At the end of Conclusions is the strange sentence:

“This section is not mandatory but can be added to the manuscript if the discussion is 419 unusually long or complex.” – I suppose that is just some editorial error?

AUTHORS: Deleted.

REVIEWER 3: Why Appendix B is divided into 2 parts?

AUTHORS: Both appendixes are divided into two parts. Now we have reduced both font and table sizes in order to show them in only one page.

REVIEWER 3: Walter & Box 1976 cited in Table S1 is missing in references. In manuscript is another reference: Fernández, M.H.; Morales, J. Biomic specialization and speciation rates.

AUTHORS: Added.

REVIEWER 3: There are other classifications of biomes, with some more recent and including more units. Authors should justify why they decide to follow that old classification?

We used the classification by Walter (1970) for three main reasons:

1) Specialization measurement. WWF's ecoregions classification system defines 14 major habitat types, which cannot be exactly considered biomes. For example, several types mix together tropical and subtropical environments with very different climatic conditions (tropical and subtropical moist broadleaf forests). In other cases, montane ecosystems (tropical and subtropical coniferous forests, montane grasslands) are differentiated from their lowland temperate analogues (boreal forests, temperate grasslands), although climatic values are generally similar. Furthermore, there are several types that only reflect azonal habitat differences (mostly edaphic) within a climatic zone (flooded grasslands and savannas, mangroves). Although we think that the worldwide ecoregions catalogue is probably one of the most complete and precise ones for the description of regional ecosystem diversity, we consider it difficult to use in the development of an ecological specialization measure.

2) Simplicity. Walter’s (1970) classification show a more reduced number of biomes in concordance with traditionally recognized major terrestrial biomes on Earth. The global biome classification by Heinrich Walter is based on the interaction between monthly precipitation and temperature average values. It was established on the basis of the concordance among annual climate values, global ocean-atmospheric circulation patterns and regional vegetation (biome). This simplicity and its concordance with traditionally recognized major terrestrial biomes on Earth have been the reasons why this classification was originally chosen for a measure of ecological specialization (Hernández Fernández & Vrba, 2005a). We like this classification because there are only ten different climate zones, placed in only one hierarchic level, allowing to differentiate the main biomes in the world. On the contrary, most other climatic classifications differentiate many more biomes, which may hamper the statistical power of any analysis, or very few, which does not allow to properly differentiate species ecology within such loosely defined biomes.

3) BSI. The Biomic specialisation index (BSI) is the measure we work with because it is proven to be a good index to determine the degree of environmental (habitat, ecosystem, landscape, etc) heterogeneity and ecological specialisation at a global scale (Hernández Férnandez & Vrba 2005a,b c, Hernández Férnandez & Vrba 2005 Evol. Ecol., Moreno Bofarull et al. 2008, Cantalapiedra et al. 2011, Gómez Cano et al. 2013, Hernández Fernandez et al. 2015 Palaeobiol Palaeoenv, Menéndez et al. 2021 Mammal Review). This index is estimated using the Walter’s classification of biomes.

In the methods of the new version, this choice of biome classification has been better justified.

Round 2

Reviewer 3 Report

I have read again the manuscript titled “Trophic niche breadth of Falconidae species predicts biomic specialisation but not range size” (ID: biology-1610510), which was revised by Fargallo et al. Unfortunately, I am not satisfied with the revision, and responses, as Authors mostly denied to my previous comments, and change their study on a little. I hold most of my previous doubts about inequality of data in use (much more exemplary studies for Falcon and Caracara (but it is only one species), some for Daptrius, and minor for other genera), about neglecting of the species evolutionary history and phylogeographic origin, and about use of data from various periods for different taxa. I suppose that it would be better to restrict this study to only Falcon, to breeding (or sedentary) range, and consider origin of species in analyses. Finally, some parts of this paper are still too long.

Author Response

These are our responses in relation to the comments made by reviewer #3:

  1. I hold most of my previous doubts about inequality of data in use (much more exemplary studies for Falcon and Caracara (but it is only one species), some for Daptrius, and minor for other genera), (…)

We do not consider adequate to remove species from the analyses (and limit the study to the Falcon genus, as suggested below), because the evolutionary relevance of the study would be affected. It is not easy to have feeding studies for 61 wild species of a given taxon. We consider this fact as a great opportunity instead of a great problem. We can think so because we trust the statistical tools used to mitigate the potential effect of the number of studies on the variables of diet breadth. In this way, we used residuals from the relationship between trophic breadth variables and number of studies as it is explained in the manuscript. The sampling effort effect was observed in DR and MDD, but not in DD. The increase in richness and maximum diversity as the number of studied populations do is controlled and detrended to make the studies comparable. This is a correct way to proceed and it is clearly exposed in our study.

  1. (…) about neglecting of the species evolutionary history and phylogeographic origin,

We must take into account that this idea is not the only proposal to explain the latitudinal gradient in the number of species nor to explain the geographical range size and nor to explain the expansion of the species. Range size (which is not always correlated with latitude, not even in birds) and species expansion has been explained through ecological factors, such as body size, wing length (in birds and invertebrates), dispersal capacity, migratory status, climatic tolerance (Rapoport’s “Rule”), etc. Also range size and species expansion have been explained through environmental and geographic factors, such as, latitude, longitude, elevation, continental width, temperature, biome area, etc. It is totally beyond our purpose to analyse each one of the ecological and environmental variables that could explain the range size, the environmental heterogeneity and their interactions with diet breath. This is the reason why we did not consider migratory status at the beginning of the study, since this feature is one of many others that can explain the range size or the BSI. We included it at the request of the reviewer, but it doesn't make much sense for our purpose.   

All the statistical analyses in this study have been made within an explicit phylogenetic context, as it is explained in the manuscript. Therefore, multiple factors that may have influence on the evolutionary history of these taxa are statistically taken into account, although there is no explicit mention of them. To explicitly include too many co-variables in the analyses would difficult computation and there is a limit to the detail these numerical models are able to assume.

Our objective was not to develop a complex model that includes all the variables that may influence on trophic niche breadth but to establish if there is a relationship between trophic amplitude and geographical range size or habitat occupation capability.

The experience of working in this other study propelled us to avoid here computation limitations associated to this kind of extremely complex analyses, since it was far away from our objectives in the present work.

About the idea proposed.

If we understand correctly what the reviewer suggests is that those species that expand towards higher latitudes from the sub/tropics (recent expansion), have a larger range size because they have occupied the larger biomes in the high latitudes and also have a more recent evolutionary history so they have had less time to partition their niche.

If this were so, we should have found 1) a significant phylogenetic signal in range size since certain genera are the ones that have expanded the most towards the high latitudes showing larger areas, such as Falco, while others, such as Daptrius or Micrastur, have expanded less showing smaller areas in the tropics-subtropics, and 2) that more recently expanded species occupying larger areas in larger biomes should have less environmental heterogeneity in their distribution, since they have had less time to partition their niche. Neither of the two things is corroborated by our results, since we have not found a high phylogenetic signal (in fact it is practically null) and also the range size is positively correlated with environmental heterogeneity (BSI).

Furthermore, it has been reported that it is the oldest taxa, not the most recent, that have spread the most to high latitudes (e.g. Goldberg et al. 2005 Am Nat; Roy et al. 2009 Proc R Soc Lond) or also that there is no correlation between species’ age and range size (e.g. Swaegers et al. 2014 J. Evol Biol), which suggests that it is not a general phenomenon.

In any case, as we have said above, trying to explain the effect of all the factors that could affect the range size or the environmental heterogeneity in relation to the diet breadth diet is beyond our objective in this study.

  1. (…) and about use of data from various periods for different taxa.

We do not consider appropriate to remove diet data providing from different periods because these are integral parts of the species range distributions. To remove these data would suppose to obviate relevant information for the understanding of the trophic niche of the studied species. In addition, it is expected the species to change diet in different seasons, but we cannot think of a reason why the richness or diversity of the diet should be different in different seasons. As said above, we consider that being able to have specific feeding studies is a good chance to undertake this type of study in a bird taxon. It would also be optimal if all the studies were carried out in the same period, but this is not feasible. This issue could have some uncontrolled effect on range size, but our results are conservative for BSI. Everything is explained in the study, so any reader will be able to draw their conclusions and perhaps make improvements for future studies, as is the case with all published studies.

  1. I suppose that it would be better to restrict this study to only Falcon, to breeding (or sedentary) range, and consider origin of species in analyses.

See previous explanations

  1. Finally, some parts of this paper are still too long.

In his/her previous review, this reviewer made specific reference to the introduction. Therefore, we have made and additional effort in reducing it. The new version that we present here has been reduced to a 66% of their original extension in the first draft. Additional reduction would be, in our opinion, in detriment of the establishment of an adequate reasoning sequence that allows the reader to fully understand the relevance of the aims of this research.

Other parts of the manuscript that may appear now as longer than in the original version are the product of taking into account comment by reviewers #1 and #2.